# Search for Associations of *FSHR*, *INHA*, *INHAB*, *PRL*, *TNP2* and *SPEF2* Genes Polymorphisms with Semen Quality in Russian Holstein Bulls (Pilot Study)

**DOI:** 10.3390/ani11102882

**Published:** 2021-10-02

**Authors:** Elena Nikitkina, Anna Krutikova, Artem Musidray, Kirill Plemyashov

**Affiliations:** Russian Research Institute of Farm Animal Genetics and Breeding—Branch of the L.K. Ernst Federal Research Center for Animal Husbandry, 55A, Moskovskoye sh., 196625 Saint-Petersburg, Russia; anntim2575@mail.ru (A.K.); 13linereg@mail.ru (A.M.); kirill060674@mail.ru (K.P.)

**Keywords:** sperm quality, bulls, candidate genes, polymorphism

## Abstract

**Simple Summary:**

It is important to know the semen quality of sires as quickly as possible. The development of DNA testing methods led to their active introduction into the practice of breeding farm animals. Many studies show that variants of single nucleotide polymorphism loci can be effectively used in as genetic markers in breeding. The aim of our study was to look for the association of polymorphism genes with fresh sperm quality in Russian Holstein bulls. In this pilot study we found that some genotypes of the follicle-stimulating hormone receptor gene are associated with doublet volume, concentration, and the total number of spermatozoa and some genotypes of hormone inhibin gene with higher sperm concentration and volume of ejaculate. Polymorphisms in hormone receptor genes may be good markers of spermatogenesis. This will allow for the finding of bulls with poor sperm quality at an early age.

**Abstract:**

The aim of the study was to search for new mutations in the previously studied gene loci of follicle-stimulating hormone receptor (*FSHR*), inhibin α (*INHA*), inhibin β A (*INHAB*), prolactin (*PRL*), transition protein 2 (*TNP2*), and sperm flagella 2 (*SPEF2*) by sequencing, as well as the search for associations of previously identified mutations at these loci with fresh semen quality in Russian Holstein bulls. Phenotypic data from 189 bulls was collected. Data was analyzed for most bulls for three years of semen collection. The maximum value of each semen quality indicator (doublet ejaculate volume, sperm concentration, progressive motility and total number of spermatozoa) were selected. SNPs were identified in the *FSHR*, *INHA*, *INHAB*, *TNP2*, *SPEF2* genes. The *PRL* gene did not have polymorphism. Significant (*p* < 0.05) associations of polymorphisms in the *FSHR* gene with double ejaculate volume, concentration and total number of spermatozoa were identified. Polymorphism in the *INHA* gene was significantly associated (*p* < 0.05) with sperm concentration. Polymorphism in the *INHAB* gene was significantly associated (*p* < 0.05) with doublet ejaculate volume and total number of spermatozoa. Polymorphisms in the *TNP2* and *SPEF2* genes did not have significant associations with semen quality. The SNPs studied in our pilot work may be considered as candidate genetic markers in the selection of bulls.

## 1. Introduction

The widespread use of artificial insemination in cattle breeding implies the need to select bulls based on semen quality [1,2,3]. In the Russian Federation, bulls from the best parents are selected for breeding stations at the age of 6–10 months [4]. Sometimes bulls have poor semen quality and sperm do not freeze or freeze poorly. Maintaining such bulls at the breeding station is not profitable. Therefore, it is important to look for genetic markers of semen quality in bulls in order to detect such males at an early age [5] before they enter the breeding station.

Hormones have a decisive influence on the formation and functioning of the reproductive system of animals, both in females and males [6,7,8]. Hormone receptor gene polymorphisms may be good markers of aspects of spermatogenesis. One of the most significant hormones is follicle-stimulating hormone (*FSH*), which supports the process of spermatogenesis in bulls [9]. It has been shown that inactivating mutations in the genes of follicle-stimulating hormone (*FSH*) and its receptor (*FSHR*) are causative factors in the occurrence of azoospermia, oligozoospermia and subfertility [10]. A mutation in the *FSHR* gene, leading to its inactivation, leads to varying degrees of spermatogenic insufficiency in males, but does not entail complete infertility [11]. Knockout of the *FSHR* gene in mice affects the critical phases of spermatogenesis in males, as their spermatozoa appear dysfunctional, which entails a significant decrease in fertility [12].

The hormone inhibin α (*INHA*) is a marker of the Sertoli cell function and exocrine testicular function (spermatogenesis state). Studies in humans have shown that in 100% of patients with a sperm concentration less than 20 million/mL, the serum inhibin concentration was below 80.0 pg/mL at a rate 480.0 pg/mL [13]. Studies of the influence of hormonal factors on semen quality in bulls showed that the plasma inhibin concentration in the group of bulls with abnormal sperm (morphological defects of sperm, low motility) was lower than in the group with normal sperm (*p* < 0.05) during puberty (at 5, 8–13, 16, 17, 19 and 20 months) [14]. Inhibin β A (*INHBA*), as well as inhibin α (*INHA*), affect the development of spermatogonia, the production of testosterone by Leydig cells, and modulate the pituitary follicle-stimulating hormone [15]. According to the researchers, the genes encoding the inhibins α and β (*INHA*, *INHBA*) are candidate genes for the analysis of the association with fertility [15]. Thus, in Hanoverian stallions, SNPs and haplotypes in the inhibin β A gene *(INHBA*) are reliably associated with pregnancy rate per estrus and the embryonic and paternal component of breeding values. Mutations identified in the *INHBA* gene alter binding sites, transcription factors, and can affect the expression of INHBA [16]. 

The hormone prolactin (*PRL*) is a regulatory factor in the male reproductive system, and it affects gametogenesis in the testis [17]. Studies of human sperm have shown a direct relationship between the concentration of the hormone prolactin in the seminal plasma and motility and the concentration of sperm [18]. Prolactin has a wide range of effects in mammals. It participates in more than 300 biochemical processes throughout the entire life of the body. It is possible that the results of the analysis of the association of sperm quality parameters with polymorphism in the *PRL* gene obtained by us are associated with the effect of prolactin on spermatogenic cells, as well as on the production of testosterone by Leydig cells and accessory reproductive glands, which requires further study. Knockout of the *PRL* gene of mice causes significant changes in the reproductive neuro-endocrine function in males, affects the growth of seminal vesicles and the ventral prostate [19]. The prolactin gene (*PRL*) and its receptor (*PRLR*) are candidate genes for searching for associations with the reproductive functions of males, in particular, with sperm quality, since they affect the function of the testes and accessory gonads. SNPs in these genes are associated with fertility traits in stallions [20]. 

In addition to candidate genes indirectly involved in the formation of the reproductive system of males, research into genes directly involved in spermatogenesis may be promising. One of these is the transition protein 2 (*TNP2*) gene. Transition proteins, including the TNP2 protein, which are a group of arginine and lysine-rich proteins, replace nucleohistones, and then are replaced by protamines [21], and chromatin condensation is initiated [22]. Biochemical analyzes have shown that the TNP2 protein increases the melting temperature of DNA and promotes the formation of compact DNA packing in the nucleosome nuclei, thus TNP2 is a DNA condensing protein [23]. Knockout of the *TNP2* gene in mice confirmed its key role in the formation of sperm structures and functions. *TNP2* knockout mice had abnormalities in the structure of the sperm, in particularly with regard to abnormalities of the head and pathology of the acrosome, as well as defects in sperm motility and their inability to penetrate the zona pellucida [24]. Studies of human *TNP2* gene polymorphism have shown that SNP (rs8043625) is associated with male infertility and apparently affects the splicing processes [25].

The *SPEF2* gene (sperm flagella 2) is involved in the formation and development of male germ cells and is a key factor in sperm motility. Expression of the *SPEF2* gene was detected in germ cells, Sertoli cells, in the tails of elongated spermatids, and in the tails of mature spermatozoa [26]. A mutation in the *SPEF2* gene, expressed by the insertion of a long disseminated element 1 (L1) in intron 30 of the *SPEF2* gene is the cause of the sperm defect-ISTS-immobile spermatozoa with a shortened tail in Yorkshire boars. The increased frequency of mutation in the Finnish Yorkshire pig population is associated with a positive effect of L1 insertion into the *SPEF2* gene on litter size in the first offspring [27]. The effect of an increase in litter size upon insertion into the *SPEF2* gene is also observed in goats [28]; however, the effect of this mutation on goat sperm has not yet been studied. Mutations in the *SPEF2* gene, leading to the loss of its function, are also the cause of impaired spermatogenesis in mice [29]. Multiple morphological abnormalities of sperm flagella (MMAF) are a consequence of the expression of a defective SPEF2 protein caused by mutations in the human *SPEF2* gene [30].

The aim of the pilot study was to search for associations in the *FSHR*, *INHA*, *INHAB*, *PRL*, *TNP2* and *SPEF2* genes polymorphisms with the semen quality in Russian Holstein bulls.

## 2. Materials and Methods

### 2.1. Ethics Statement

The principles of laboratory animal care were followed, and all procedures were conducted according to the ethical guidelines of the L.K. Ernst Federal Science Center for Animal Husbandry (Protocol Number: 2020/2) and the Law of the Russia Federation on Veterinary Medicine No. 4979-1 (14 May 1993).

### 2.2. Experimental Animals and Phenotypes

Phenotypic data from 189 bulls was collected at the Open Joint Stock Company “Nevskoye”, Saint-Petersburg, Russia. We used archived data during the period 2007–2018. Bulls lived at the station from six months of age. Semen was collected at intervals of three to four days. Data was analyzed for most bulls for three years of semen collection. For bulls with poor sperm quality, data was evaluated until culling. An average of 145 (from 47 to 215) ejaculates per bull was analyzed. The maximum values of each sperm quality indicator were selected. Sperm quality indicators included semen volume per doublet ejaculate (DVOL, mL), sperm concentration (SCON × 10^6^/mL), progressive motility (PMOT, %), and total number of spermatozoa (TNS, × 10^8^). The cumulative volume of two ejaculates (DVOL) collected 20 min apart was assessed indirectly by weight immediately after collection. Sperm concentration was measured with a Photometer SDM 1^®^ (Minitüb GmbH, Tiefenbach, Germany) calibrated for bovine. Sperm concentration and progressive motility (PM) were evaluated by a computer-assisted sperm analysis (CASA) in a Mackler chamber at 37 °C. Argus CASA system (ArgusSoft LTD., St. Petersburg, Russia) and a Motic BA 410 microscope (Motic, Hong Kong, China) were used. The total number of spermatozoa was determined by multiplying DVOL by SCON.

### 2.3. Isolation of Genomic DNA

Genomic DNA was isolated from frozen semen samples from 189 animals by the phenol-chloroform method using mercaptoethanol and proteinase K. The resulting DNA was dissolved in TE buffer. The quality and quantity of the obtained DNA was evaluated on a NanoDrop 2000 spectrophotometer (Thermo Scientific, Waltham, MA, USA). The isolated DNA was stored at a temperature of −20 °C.

### 2.4. PCR Conditions

PCR was performed in 10 μL of the reaction mixture containing 2 μL of 5 × Taq buffer (15 mM Mg^2+^) (Evrogen, Moscow, Russia), 0.2 μL of Taq polymerase (Sibenzim, Novosibirsk, Russia), 1.2 μL of a mixture of dNTP (2.5 mM) (Thermo Fisher, Waltham, MA, USA), 0.4 μL of each of a pair of primers (Syntol, Moscow, Russia), 5.8 μL of bi-distilled water, and 0.4 μL of genomic DNA (100 ng/μL) as matrices. The reaction was carried out at the following temperature and time parameters: starting denaturation 95 °C for 5 min; 35 cycles: denaturation 95 °C for 20 s, annealing of primers-at a temperature specific for each pair of primers for 20 s, elongation 72 °C for 20 s; final elongation 72 °C for 5 min To analyze the sites of the selected genes, the primers used are shown in Table 1.

### 2.5. Sequencing and RFLP Reactions

A portion of each sample was sequenced on an Applied Biosystems 3500 Genetic Analyzer using the BigDye^®^ Terminator v3.1 Sequencing Standard Kit (Applied Biosystems, Foster City, CA, USA) according to the manufacturer’s protocol. Genotyping was then performed by PCR-RFLP. The mutations in the regions of the studied genes were identified using the restriction endonucleases *FSHR*-*Taq*I, *INHA*-*Msp*I, *TNP2*-*Hind*III, and *SPEF2*-*Taq*I. The PCR product was cleaved using 2 U of each restriction enzyme for 2 h at 65 °C (for *Taq*I) and 37 °C (for the rest). Fragments were detected on a 2% agarose gel with ethidium bromide. Two sections of the PRL gene were analyzed by sequencing only.

### 2.6. Statistical Analyses

Data was analysed by ANOVA using IBM SPSS Statistics 19 (IBM, Armonk, NY, USA) and Statistica 10 (TIBCO Software Inc., Palo Alto, CA, United States). The data were expressed as means ± standard deviation. All pairwise multiple comparisons between means were conducted by *t*-test. To control the type 1 error a Bonferroni test was performed. Differences were considered statistically significant at *p* < 0.05.

## 3. Results

The maximum values of the studied indicators of semen quality varied between bulls. The maximum DVOL varied from 3 to 27 mL, the maximum sperm concentration was from 700 × 10^6^/mL to 2000 × 10^6^/mL, and the total number of spermatozoa was from 2.7 × 10^8^ to 26.4 × 10^8^ in different bulls. This enabled bulls with contrasting semen quality indicators to be selected for PCR and sequencing.

Three new SNPs in exons 4 and 5 of the PRL gene discovered by L. Sang et al. (2011) could not be detected by RFLP using the restriction enzymes *Rsa*I, *Alu*I and *Aci*I, while sequencing showed the absence of these SNPs in our samples of Russian Holstein bulls.

The single nucleotide polymorphism of C7639T (rs43408735) in intron 1 of the *INHAB* gene, declared by other researchers, was not detected by us. However, at this point of this locus, according to the international genetic database Ensembl and according to the results of our sequencing, there is an SNP identified as A > G (rs43408735) [31]. This polymorphism is not detected by the *Taq*I endonuclease since none of the polymorphic variants forms a restriction site.

Polymorphism in *FSHR* (Figure 1), *TNP2* (Figure 2), *SPEF2* (Figure 3), and *INHA* (Figure 4)*, INHAB* genes was found. The frequency of genotypes and alleles were calculated by direct counting (Table 2). TT genotype of *FSHR* gene was rare. CC and CT genotypes in *TNP2* gene were also rare. AA genotype in *INHA* gene was not found.

Descriptive statistics for the semen quality traits are listed in Table 3. Only the maximum values of the traits for each bull are included in the analysis. From the data in the table it can be seen that the quality of the sperm was different.

The results of association analysis between *FSHR*, *INHA*, *INHAB*, *TNP2* and *SPEF2* genotypes and sperm quality traits are shown in Table 4. Bulls with the AA (*FSHR*) (rs43676359) genotype had a significantly (*p* < 0.05) higher doublet volume, concentration, and the total number of spermatozoa than did those with the TT genotype.

Bulls with the genotype GG (*INHA*) (rs41257116) had a higher cell concentration than did bulls with the genotype AG (*p* < 0.05). No significant differences in ejaculate volume and motility were found. However, bulls with the GG genotype had greater ejaculate volume and higher motility than bulls with the AG genotype.

Bulls with the genotype AA (*INHAB*) had the highest doublet ejaculate volume (19.24 ± 6.35 mL) and total number of spermatozoa (32.54 ± 8.59 mln) compared to bulls with the genotypes AG (16.45 ± 8.67 mL and 23.35 ± 4.32 mln) and GG (11.32 ± 7.57 mL and 14.85 ± 1.98 mln) (*p* < 0.05). No significant differences were found in the concentration of spermatozoa and motility. However, bulls with the AA genotype had the greatest concentration and motility. The bulls with the GG genotype had the poorest semen quality.

There were no significant differences for all studied parameters in bulls with different genotypes for the *TNP2* and *SPEF2* genes.

## 4. Discussion

In our study, we did not take into account many other factors, especially the age of the bull and the season of the year. An average of 145 (from 47 to 215) ejaculates per bull was analyzed. The analysis took the maximum values of sperm quality traits, which were for three years of semen collection. Therefore, we decided not to include the other factors in the analysis. The bulls were all in the same condition. 

Polymorphism in two sections of the PRL gene was not detected in our study. How-ever, the studies of the Chinese population of Holstein bulls showed the presence of three SNPs in these areas. Two mutations were identified, one in exon 4 (A7550G and C7661T) and one in exon 5 (T8370C). Despite the presence of a high level of polymorphism in the three SNPs, no statistically significant effect on sperm quality was detected [32].

Our data partly coincides with the Chinese Holstein data with respect to the influence of the *FSHR* gene on semen quality. In their studies, Holstein bulls with AA genotype also had significantly (*p* < 0.05) higher ejaculate volumes and concentrations [32], although they did not take the total number of spermatozoa into account. There were no significant differences between heterozygotes and homozygotes in our work. However, semen quality was better in bulls with the heterozygous AT genotype than in homozygous TT bulls. Such an effect of a single nucleotide substitution in the 5′-upstream region of the *FSHR* gene can occur due to a change in the level of transcriptional activity [33] as a result of changes in the transcription factor binding sites. Such changes lead to a decrease in the level of *FSHR* gene expression, which, in turn, affects spermatogenesis. Further research is needed involving a larger number of bulls, especially ones with the TT genotype.

The *INHA* gene is associated with histological maturation of the testes and, thus, has a significant effect on bovine fertility [34]. Thus, variants of polymorphism in the *INHA* gene could be candidate mutations for sperm quality. In addition, the A192G SNP in the *INHA* gene is associated with spermatozoa respiration [32]. The mutation in the *INHA* gene revealed by us in Russian Holstein bulls is synonymous, which does not lead to amino acid replacement. Despite this, however, it is believed that the mutation affects the expression of the *INHA* gene and the stability of *INHA* transcription, which leads to a decrease in the concentration of inhibin. This leads to an increase in FSH concentration due to impaired feedback from the pituitary gland [35]. Also, a synonymous nucleotide substitution can affect the proper formation of protein structures during biosynthesis, which, accordingly, will affect gene expression, protein function and phenotype [36]. In our studies, bulls with the AA genotype were not identified. Bulls with the genotype GG had a higher sperm concentration than bulls with the genotype AG (*p* ˂ 0.05). No significant differences in ejaculate volume and motility were found. Obviously, differences in sperm quality are associated with the effect of mutations on the regulatory functions of inhibin on cell proliferation during spermatogenesis. This is confirmed by studies showing that the level of inhibin positively correlates with sperm concentration [37]. The lack of statistical certainty is probably associated with a small number of individuals with a certain genotype.

Using immunohistochemical analysis, *INHAB* was detected in Sertoli cells, Leydig cells, and testicular endothelial cells of newborn rats three to nine days old [38]. *INHAB* has been associated with the proliferation of immature Sertoli cells and the differentiation of spermatogonia [38,39,40]. The *INHAB* protein can regulate the processes of gametogenesis in immature and adult testes, and also affect Sertoli cells [41]. For male pigs, polymorphisms in the *INHAB* gene are markers of boar fertility, since they are associated with sperm plasma quality and sperm rate. As a result of studying the *INHAB* gene by Chinese scientists, four SNPs were found (C7639T (rs43408735), G7550A, C7661T, T8370C), three of which were first identified. However, only one of these C7639T polymorphisms, according to the authors, had a significant relationship with ejaculate volume, sperm concentration and motility (*p* < 0.05) [32]. In our studies, the same region of the *INHAB* gene was sequenced, as a result of which only one nucleotide substitution A > G (rs43408735) was identified. The substitution is located in intron 1 and does not lead to a change in the amino acid composition of the protein; however, the proximity to the exon can affect the splicing process. The frequency of the alleles in our studies was A—0.55, G—0.45, which differs from the population data in the ENSEMBL database, where the frequency of alleles was A—0.31, G—0.69. The frequency of occurrence of genotypes according to our data was AA—0.35, AG—0.40, GG—0.25. 

The TNP2 protein is a transitional protein and is involved in the removal of histones and chromatin condensation during sperm formation and maturation [42]. Gao Q. et al. (2014) revealed that bulls with the CT genotype in the *TNP2* gene are characterized by higher volumes of ejaculate and more motile sperm after thawing than in bulls with CC genotype, as the T > C mutation in the 3′-UTR region affects the expression of *TNR2* mRNA, which determines significant differences in sperm quality. Our studies confirm this work in that bulls with the CC genotype had the lowest ejaculate volume and motility compared with the other genotypes. However, differences in our study were not significant. Bulls with the TT genotype had the highest semen quality (VOL, SCON and TNS) in contrast to the results of Gao et al. [42]. In their studies, bulls with the CT genotype had the best semen qualities. In our studies, there were no significant differences in bulls with different genotypes for the *TNP2* gene, which may be due to the small number of animals involved. 

The *SPEF2* gene plays a significant role in the formation and development of the sperm tail. Mutations in this gene lead to impaired spermatogenesis, the appearance of tail defects and, accordingly, impaired sperm motility [29]. Mutations in the *SPEF2* gene, which has a complex structure and consists of 36 exons and 35 introns, leads to alternative splicing and, as a result, the loss of protein functions that play a role in spermatogenesis and testicular development, leading to sperm immobility [43]. In our studies, there were no significant differences in semen quality in bulls with different genotypes for the *SPEF2* gene. However, bulls with the heterozygous genotype TG had higher values for all the studied parameters than those with the homozygotes GG and TT. The single nucleotide substitution in the *SPEF2* gene leads to alternative splicing and, as a result, to two alternative variants of the *SPEF2* protein. A combination of two alternative proteins probably contributes to a more efficient formation of sperm motility factors during spermatogenesis. It has been noted that bulls with the GG genotype had the greatest sperm motility and fewer defects [44]. However, as these studies were conducted on frozen-thawed semen it is possible that this polymorphism also affects the preservation of sperm during cryopreservation. At present, the molecular mechanisms of regulation of this process by the *SPEF2* protein remain unknown [44].

## 5. Conclusions

When analyzing the results of our pilot studies on the search for associations between single nucleotide substitutions in the genes responsible for the reproductive function of bulls, it is necessary to take into account that prolonged use of artificial insemination contributes to the natural elimination of undesirable infertility genotypes from the population. This process is due to the selection of bulls with the best semen quality for further reproduction. Thus, the frequency of occurrence of genotypes unfavorable for the qualitative characteristics of sperm is significantly reduced, which can affect the results of statistical analysis in such studies. However, semen from bulls of outstanding genetic production potential, with albeit insufficiently high genetically determined indicators of semen quality, is undoubtedly so widely used as to perpetuate valuable genetic material. Therefore, genotypes undesirable for semen quality are found in the population. The SNPs studied in our work in genes that affect the reproductive function of bulls can be considered as a candidate for its use in identifying molecular genetic markers in the selection of young bulls. The study of the molecular basis of the influence of these mutations should allow a deeper understanding of the principles of the formation of fertility in bulls.

## Figures and Tables

**Figure 1 animals-11-02882-f001:**
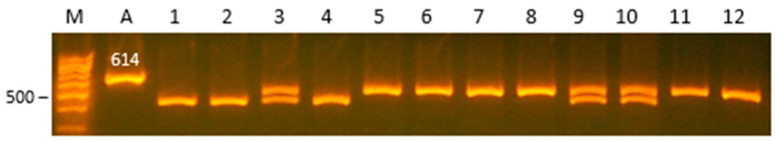
The *Taq*I polymorphism g.234500 A > T in gene *FSHR*: 1, 2, 4—genotype AA (446, 97, 71 bp);3, 9, 10—genotype AT (517, 446, 97, 71 bp); 5–8, 11, 12—genotype TT (517, 97 bp); A—amplicon; M—marker.

**Figure 2 animals-11-02882-f002:**
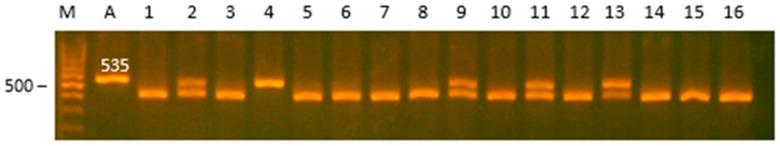
The *Hind*III polymorphism g.1536 C > T in gene *TNP2*: 1, 3, 5–8, 10, 12, 14–16—genotype TT (432, 103 bp); 2, 9, 11, 13—genotype CT (535, 432, 103 bp); 4—genotype CC (535 bp); A—amplicon; M—marker.

**Figure 3 animals-11-02882-f003:**
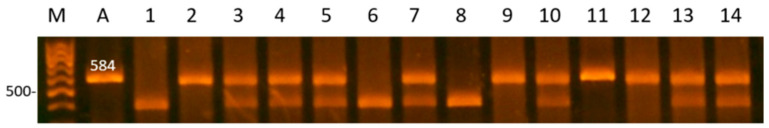
The *Taq*I polymorphism c.2851 G > T in gene *SPEF2*: 2, 9, 11, 12—genotype TT (584 bp); 3–5, 7, 10, 13, 14—genotype GT (584, 460, 124 bp); 1, 6, 8—genotype GG (460, 124 bp); A—amplicon; M—marker.

**Figure 4 animals-11-02882-f004:**
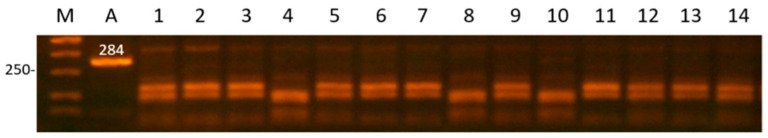
The *Msp*I polymorphism c.192 A > G in gene *IHNA*: 1–3, 5-7, 9, 11–14—genotype AG (123, 95, 79, 44, 31 bp); 4, 8, 10—genotype GG (95, 79, 44, 31bp); A—amplicon; M—marker.

**Table 1 animals-11-02882-t001:** The list of primers used in *FSHR*, *INHA*, *INHAB*, *PRL* (Sang et al., 2011), *TNP2* (Gao et al., 2014) and *SPEF2* (Guo et al., 2014) genes.

Gene	Locus	Primer Sequence (5′ > 3′)	AT (C^0^)	RE	RES, bp/Genotype
*FSHR*	5′-upstream region	F:TGTTCCCACTGACTCAGATACTT R:TCCCTGCCCTTCAGTGACGAA	61	*Taq*I	AA: 446, 97, 71 AT: 517, 446, 97, 71 TT: 517, 97
*INHA*	Part of exon 1	F:ATTCAACCCAACCTGCCTA R:GCCCTGTTTCTGGATGCC	61	*Msp*I	AA: 123, 95, 31 AG: 123, 95, 79, 44, 31 GG: 95, 79, 44, 31
*INHAB*	Intron 1	F: CAGGGTTTCAGAAGTTGG R:GGTGGTTGTTACTGTTTATC	61	Seq ^1^	
*PRL-1*	Complete of exon 4	F:GAGATTGTTTCTTGTGGTTGTTCAG R:TTCAAAACTCATTCCTCATTACAGA	62	*Rsa*I	284 n/p ^2^
*PRL-2*	Part of exon 5	F:GATAAATAGAAAGAACAAAGATGAG R:AACCCATTAGAGCCAAGC	60	*Alu*I	372 n/p ^2^
*TNP2*	3′-upstream region	F:ACTGGACCAATGAACGAA R:CTCCCTACCCAACCTCTT	61	*Hind*III	TT: 432, 103 TC: 535, 432, 103 CC: 535
*SPEF2*	5′-upstream region	F: TAATCCTCCACCAGAATCTG R:GCCTTATGATGTGGGTATGA	61	*Taq*I	TT: 584 TG: 584, 460, 124 GG: 460, 124

AT—annealing temperature of primers; RE—restriction enzyme; RES—restriction fragment size. ^1^—sequence only. ^2^—not polymorphic. bp—base pairs.

**Table 2 animals-11-02882-t002:** Frequency of genotypes and alleles of *FSHR*, *INHA*, *INHAB, TNP2* and *SPEF2* genes in Russian Holstein bulls.

Gene	Genotype	Frequency of Genotypes	Alleles
*FSHR* (rs43676359)	AA (69)	0.365	A-0.651
	AT (108)	0.571	T-0.349
	TT (12)	0.063	
*INHA* (rs41257116)	AA (0)	0.000	A-0.413
	AG (156)	0.825	G-0.587
	GG (33)	0.175	
*INHAB* (rs43408735)	AA (69)	0.365	A-0.569
	AG (77)	0.407	G-0.431
	GG (43)	0.227	
*TNP2* (g.1536)	CC (21)	0.111	C-0.196
	CT (32)	0.169	T-0.804
	TT (136)	0.720	
*SPEF2* (c.2851G>T)	GG (65)	0.344	G-0.603
	GT (98)	0.518	T-0.397
	TT (26)	0.138	

Note: Figure in brackets are the number of samples.

**Table 3 animals-11-02882-t003:** Descriptive statistics for the semen quality traits.

Gene	DVOL (mL)	SCON (×10^8^/mL)	TNS (×10^8^)	PMOT (%)
Number of samples	189	189	189	189
means	16.3	1.45	24.16	85.9
standard deviation	5.0	0.22	13.5	4.9
minimum	3	0.70	2.7	80.0
maximum	27	2.00	46.0	90.0

Note: semen volume per doublet ejaculate (DVOL, mL), sperm concentration (SCON ×108/mL), progressive motility (PMOT, %), the total number of spermatozoa (TNS, ×108). Figure in brackets is the number of samples. The dates are expressed as means ± standard deviation.

**Table 4 animals-11-02882-t004:** Distribution of semen quality traits by genotypes *FSHR*, *INHA*, *INHAB*, *TNP2* and *SPEF2* genes genotypes in Russian Holstein bulls.

Gene	Genotype	DVOL (mL)	SCON (×10^8^/mL)	TNS (×10^8^)	PMOT (%)
*FSHR*	AA (69)	17.2 ± 5.43 ^a^	1.41 ± 0.21 ^a^	25.8 ± 10.55 ^a^	86.3 ± 5.21
	AT (108)	16.8 ± 7.23	1.45 ± 0.27	25.2 ± 14.83	84.3 ± 4.27
	TT (12)	11.965 ± 7.58 ^b^	1.29 ± 0.32 ^b^	15.6 ± 12.16 ^b^	82.9 ± 5.13
*INHA*	AG (156)	16.56 ± 5.67	1.39 ± 0.26 ^a^	23.55 ± 12.54	84.24 ± 4.67
	GG (33)	17.20 ± 6.83	1.71 ± 0.29 ^b^	28.43 ± 13.76	86.76 ± 5.59
*INHAB*	AA (69)	19.24 ± 6.35 ^a^	1.45 ± 0.22	32.54 ± 8.59 ^a^	87.13 ± 4.43
	AG (77)	16.45 ± 8.67 ^b^	1.46 ± 0.33	23.35 ± 4.32 ^b^	83.21 ± 5.24
	GG (43)	11.32 ± 7.57 ^b^	1.30 ± 0.31	14.85 ± 1.98 ^b^	83.56 ± 5.37
*TNP2*	CC (21)	14.70 ± 8.65	1.35 ± 0.36	19.64 ± 16.93	83.48 ± 5.56
	CT (32)	14.50 ± 9.72	1.26 ± 0.36	19.29 ± 16.06	86.23 ± 5.49
	TT (136)	16.21 ± 7.74	1.43 ± 0.24	25.37 ± 12.23	85.57 ± 5.34
*SPEF2*	GG (65)	16.34 ± 7.87	1.41 ± 0.23	23.33 ± 12.74	84.83 ± 5.23
	GT (98)	17.00 ± 7.89	1.43 ± 0.23	25.84 ± 12.20	85.30 ± 4.89
	TT (26)	15.46 ± 9.64	1.38 ± 0.42	23.38 ± 17.99	85.12 ± 5.49

Note: semen volume per doublet ejaculate (DVOL, mL), sperm concentration (SCON ×10^8^/mL), progressive motility (PMOT, %), the total number of spermatozoa (TNS, ×10^8^). Figure in brackets is the number of samples. The dates are expressed as means ± standard deviation. ^ab^—*p* < 0.05.

## Data Availability

Data will be made accessible from corresponding authors upon reasonable request.

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
