# Peer review of "Search for Associations of *FSHR*, *INHA*, *INHAB*, *PRL*, *TNP2* and *SPEF2* Genes Polymorphisms with Semen Quality in Russian Holstein Bulls (Pilot Study)"

_animals, 2021, doi:10.3390/ani11102882_

Round 1

Reviewer 1 Report

It is suggested that the authors provide more descriptions of phenotypic data. Because they  used archived data during the period 2007 -2018, I cannot assess whether other fixed effects or covariates need to be added for association study without more descriptions.

I also suggest the authors perform multiple comparison tests to control the type 1 error, for example Bonferroni  or  DUNCAN methods. 

Author Response

Thank you for review our study. The answers are: 

This paper has several weaknesses and needs improvement before publication. This manuscript has major language problems. Authors are strongly encouraged to seek a native English speaker who may assist you in modifying the document.

It is suggested that the authors provide more descriptions of phenotypic data. Because they  used archived data during the period 2007 -2018, I cannot assess whether other fixed effects or covariates need to be added for association study without more descriptions.

Answer: For the evaluation, we take the maximum value for each characteristic of semen quality collected in three years bull using. The bulls were all in one place. We think the phenotypic data is sufficient for this pilot study.

I also suggest the authors perform multiple comparison tests to control the type 1 error, for example Bonferroni  or  DUNCAN methods.

Answer: Yes, we performed Bonferroni test. The information is added to the manuscript Line 172

Reviewer 2 Report

This paper has several weaknesses and needs improvement before publication. This manuscript has major language problems. Authors are strongly encouraged to seek a native English speaker who may assist you in modifying the document.

Line 21- „in-hibin” change to inhibin

Line 21 name of the genes should be written in italics

Line 51 - names of genes should be written in italics. Please correct it in the whole manuscript.

Line 73- „gametogenesis in the testes” correct to „ gametogenesis in the testis”

Line 88. Missing dot at the end of the sentence. Please check the whole manuscript.

In the whole manuscript appears a small error eg. a Lack of „.” Or double space. Please correct it in the whole manuscript.

Line 91- the name of the genes should be written in italics.

Line 138 – It is worth to add „ Genomic DNA was isolated from frozen semen samples from 189 animals by the phenol-chloroform method using mercaptoethanol and proteinase K”

Line 182 „The single nucleotide polymorphism of SNP C7639T (rs43408735)” please delete „ of SNP”

Line 186 – link does not work.

Line 147.  „1, 2 ul of mixture” please use  „.”  „1.2ul of mixture”

Table 1. extend the abbreviations „AT”, „RE”, „RES”.

Table 2. Please add information on how to calculate frequency alleles and genotypes? Did the population was in HWE?

Table2. IN spef2 missing „ )”

Table 3. Please explain the small letters „a” and „b”. Maybe you used a post hoc test eg. Duncan? If yes, please add appropriate information to the manuscript.  Did you including age and season in the statistical model?

Table 4. In the FSHR gene in row „genotype” please correct genotype „A” to „AA”.

Under table 4 please add information „semen volume per doublet ejaculate 127 (DVOL, ml), sperm concentration (SCON ×106/ml), progressive motility (PMOT, %), the total number of spermatozoa (TNS, ×108).”

Line 251- such an effect could be made by the small number of animals in genotype TT.

Line 260 – please delete „ (Sang et al. 2011).”

Author Response

Thank you for review our manuscript. 

Line 21- „in-hibin” change to inhibin

Line 21 name of the genes should be written in italics

Line 51 - names of genes should be written in italics. Please correct it in the whole manuscript.

Line 73- „gametogenesis in the testes” correct to „ gametogenesis in the testis”

Line 88. Missing dot at the end of the sentence. Please check the whole manuscript.

In the whole manuscript appears a small error eg. a Lack of „.” Or double space. Please correct it in the whole manuscript.

Line 91- the name of the genes should be written in italics.

Line 182 „The single nucleotide polymorphism of SNP C7639T (rs43408735)” please delete „ of SNP”

Line 147.  „1, 2 ul of mixture” please use  „.”  „1.2ul of mixture”

Table2. IN spef2 missing „ )”

Table 4. In the FSHR gene in row „genotype” please correct genotype „A” to „AA”.

Line 260 – please delete „ (Sang et al. 2011).”

Answer: We have done all recommended changes and corrected the whole manuscript.

Line 138 – It is worth to add „ Genomic DNA was isolated from frozen semen samples from 189 animals by the phenol-chloroform method using mercaptoethanol and proteinase K”

Answer: We added „ Genomic DNA was isolated from frozen semen samples from 189 animals by the phenol-chloroform method using mercaptoethanol and proteinase K”.

Line 186 – link does not work.

Answer: We checked the link. Now it is works

Table 1. extend the abbreviations „AT”, „RE”, „RES”.

Answer: We extended the abbreviations „AT”, „RE”, „RES”. Information is added to manuscript.

Table 2. Please add information on how to calculate frequency alleles and genotypes? Did the population was in HWE?

Answer: Frequency A - p, frequency a - q, p + q = 1, genotype frequencies were calculated using the following formula: (p + q) 2 = p2 (AA) + 2pq (Aa) + q2 (aa). The Hardy-Weinberg equilibrium was not calculated, since the sample does not reflect the structure of the population, since the sample is not representative as cows are not included in the analysis.

Table 3. Please explain the small letters „a” and „b”. Maybe you used a post hoc test eg. Duncan? If yes, please add appropriate information to the manuscript.  Did you including age and season in the statistical model?

Answer: The small letters „a” and „b” mean a significant difference between traits at P < 0.05. All pairwise multiple comparison between means were conducted by T-test. To control the type 1 error Bonferroni test was performed. Information is added to the manuscript. We did not include age and season in the statistical model. For the evaluation, we take the maximum value for each characteristic of semen quality obtained in three years bull using.

Under table 4 please add information „semen volume per doublet ejaculate 127 (DVOL, ml), sperm concentration (SCON ×106/ml), progressive motility (PMOT, %), the total number of spermatozoa (TNS, ×108).”

Answer: We added information: semen volume per doublet ejaculate (DVOL, ml), sperm concentration (SCON ×108/ml), progressive motility (PMOT, %), the total number of spermatozoa (TNS, ×108).

Line 251- such an effect could be made by the small number of animals in genotype TT.

Answer: Yes, we agree that such an effect could be made by the small number of animals in genotype TT.

Reviewer 3 Report

Dear authors

Please see comments on the file attached. Please carefully read your work for language errors.

Check the subscripts  and Figures 3 & 4. Not clear.

regards

Author Response

Thank you for review our manuscript. Answers:

Line 12 – we rephrased the statement to “Many studies show that variants of single nucleotide polymorphism loci can be effectively used in breeding as genetic markers”.

Line 16, 26,32,41 and 113 – we do not understand the remarks

Line 42-   Bulls are at semen collection stations from 10 months of age. Money is spent on them. Some young bulls have low semen quality. But before they can cull they are on the station up to 2 years. So Maintaining such bulls at the breeding station is not profitable.

Line 66 – We added reference

Line 76 – We have rephrased “Perhaps this is due to the direct effect of  prolactin on spermatogenic cells, as well as on testosterone production by Leydig cells” to “Prolactin has a wide range of effects in mammals. He participates in more than 300 biochemical processes throughout the entire life of the body. It is possible that the results of the analysis of the association of sperm quality parameters with polymorphism in the PRL gene obtained by us are associated with the effect of prolactin on spermatogenic cells, as well as on the production of testosterone by Leydig cells and accessory reproductive glands, which requires further study”.

Line 141- We added reference to protocol. The acceptable quality parameters are in the reference to protocol.

Line 174 - These are the maximum values of the indicators of the quality of semen of bulls.

Line 177 - we removed this sentence

Line 184 – Sang et.al, 2011  writes that at this locus there is a substitution of nucleotide C (cytosine) for T (thymine), however, both in international databases and in our studies, this particular substitution in this locus with the number 43408735 is identified as a substitution of nucleotide A (adenine) for G ( guanine).

Table 3 – N is the number of samples. We added it in the table

Line 204 - The influence of genes is presented in table 4.

Line 209 - Sperm

Line 230 - We disagree, the gel is clean

Table 4 – There is no significant difference between AG and GG. We did not understand what to check

Line 242 - We have rephrased this sentence to “However, the studies of Chinese population of Holstein bulls did showed the presence of three SNPs in these areas”.

Line 251 - AT is heterozygous and TT genotype is homozygous

Line 326 - We have rephrased part of the conclusion

Round 2

Reviewer 2 Report

Dear Authors,

All my previous suggestions and suggestions from other Reviewers` have been introduced into the manuscript. 

Kind regards 

Reveiwer

Author Response

Dear reviewer,

Thank you very much for your time and  reviewing our study.

best wishes

Authors

This manuscript is a resubmission of an earlier submission. The following is a list of the peer review reports and author responses from that submission.

Round 1

Reviewer 1 Report

The authors describe associations of FSHR, INHA, INHAB, PRL, TNP2, and SPEF2 genes polymorphisms with fresh semen quality in 32 Russian Holstein bulls.

Regarding the association analysis, I have a serious concern:

L26: The sample size is very small, with only 32 bulls. This can lead to biased results. Phenotypic data from 189 bulls was already collected and genomic DNA was isolated from frozen semen samples. I would suggest that all bulls are selected for PCR and sequencing.

L129-130: Specify the analyses and models. In addition, association analysis between genes genotypes and sperm quality traits in Table 3 is not an appropriate analysis because this ignores fixed effects, including age and season when frozen semen samples were collected.

L172-175: Many studies have analyzed additive and dominance effects of genes SNPs on traits. This deserves some results and discussion.

Pinto, L. F., et al. (2010). "Association of SNPs on CAPN1 and CAST genes with tenderness in Nellore cattle." Genetics and Molecular Research 9(3): 1431-1442.

Dagnachew, B. S., et al. (2011). "Casein SNP in Norwegian goats: additive and dominance effects on milk composition and quality." Genetics Selection Evolution 43(1): 31.

Minor:

L40: “(Saksa, 2018)” was removed. Please check and correct all.

L135-139: Please add the table of descriptive statistics for the semen quality traits.

L165-167: “Figure 1. The TaqI polymorphism g.234500 A>T in gene FSHR: 1, 3, 5-8, 10, 12, 14-16 – genotype AA (446, 97, 71 bp); 3, 9, 10 – genotype AT (517, 446, 97, 71 bp); 5-8, 11, 12 – genotype TT (517, 97 bp).” The 5-8 and 12 are genotype AA and TT, please check.

Reviewer 2 Report

In their manuscript, the authors described the search for associations between genes and sperm quality in Holstein bulls.
At the very beginning, I would like to emphasize that the manuscript must be thoroughly rebuilt. The authors undoubtedly followed one of the works quoted in the manuscript, but they did not avoid errors.
Starting with an abstract:
line 21- inhibin alpha-beta. (INHAB). Later in the text inhibin BA.
the introduction is written very generally. What is striking is the non-uniform citation of works. Once [4] and the last name, once again [4]. e.g. line 40. Please check the authors' guide and correct your work according to the guidelines.
The introduction is worth writing something about mutations in these genes in cattle, but it can be mentioned in other species whether such studies have been carried out.
Line 60; inhibin BA again. standardize and improve.
Materials and methods
paragraph 2.2: please write whether the bulls came from one region or a different? Were they related to each other?
How the semen was secured after collection and briefly describe the method of semen collection and analysis.
paragraph 2.4
line 109 - reactions were carried out in 10 µl. the sum of the ingredients gives 9.8 ul.
line 111- here's a bug.
line 112 - please specify the concentration of DNA used for the PCR reaction.
Table 1 requires a major revision.
If the primer sequences come from another publication, they must be indicated in the title of the table, the table itself, or the text. If the authors have designed primers, this should be mentioned. In this work, the primer sequences are taken from other publications.
Standardize the form of recording sequences. Once is without "-" and once with "-".
Align the text in the table. Additionally, it is worth adding a column with SNP, restriction enzyme, amplicon length, and the obtained fragments and identified allele size. (similar to doi: 10.1371 / journal.pone.0084355)
paragraph 2.5. If all samples were sequenced, why digest them with enzymes if they could be read from the sequence? It is worth adding pictures of SNP chromatographs.
paragraph 2.6 the statistical analysis is very poor. the t-student test requires normal distribution. Have they been checked?
Why were bull age and environmental impact not included in the statistical analyzes? Besides, 32 bulls are not enough for association analysis.
There are two tables 2.
Table 2 (the one with genotype frequencies). Why summing up the number of individuals of different genotypes, does it get the number 32 only in INHA and 31 in the rest?
Table 2 (3). Very large deviations. A small number of individuals in a group produces the same results. More individuals should be used for the analyzes.
To be honest, after the introduction, materials, and methods, and results, I no longer had the strength to write comments about the discussion.
Dear authors. The work still requires a little bit of attention and more effort. I hope that I will get the refined work for review and I will be happy to review it. At the moment, the work has many errors and shortcomings. Maybe you should think about writing a brief note instead of an article. Your work was probably inspired by the manuscript Sang et al. 2011. See how it is written and what analyzes were made there.

best regards

Reviewer

Reviewer 3 Report

The manuscript of Nikitkina et al describes the identification of polymorphisms in FSHR, INHA, INHAB, PRL, TNP2 2 and SPEF2 genes and their association analysis with Semen Quality in Russian Holstein bulls. The manuscript presents a proper introduction. However, going through the methods and results sections, several issues can be pointed.

  1. I do not understand why only specific parts of genes (i.e. promoter, exons, etc..) have been investigated. For a precise evaluation of candidate genes, the whole gene should be investigated.
  2. After having obtained PCR products, I do not understand how SNPs were identified. Did you carry out sequencing of fragments and then you studied the RFLP protocol? A table that reports the identified polymorphisms with their info (with the position of the SNP on the reference genome or gene, SNP type, rs code if present), is mandatory.
  3. ANOVA and T-test tests were carried out. However, I do not understand If repeated measures were present for each sample (as data comes over time) and how the repetitions were handled in the statistical analysis.
  4. It is not enough to report “P<0.05”. First of all, the P should be stated for each tested SNP. It means that it should be presented in a Table.
  5. Multiple testing correction should be addressed. The manuscript reports a total of 5 variants. It means that considering alpha = 0.05, Bonferroni corrected P should be P< 0.05/5=0.01.